# DRMGuard: Defending Deep Regression Models against Backdoor Attacks

## Abstract

Deep regression models are used in a wide variety of safety-critical applications, but are vulnerable to backdoor attacks. Although many defenses have been proposed for classification models, they are ineffective as they do not consider the uniqueness of regression models. First, the outputs of regression models are continuous values instead of discretized labels. Thus, the potential infected target of a backdoored regression model has infinite possibilities, which makes it impossible to be determined by existing defenses. Second, the backdoor behavior of backdoored deep regression models is triggered by the activation values of all the neurons in the feature space, which makes it difficult to be detected and mitigated using existing defenses. To resolve these problems, we propose DRMGuard, the first defense to identify if a deep regression model in the image domain is backdoored or not. DRMGuard formulates the optimization problem for reverse engineering based on the unique output-space and feature-space characteristics of backdoored deep regression models. We conduct extensive evaluations on two regression tasks and four datasets. The results show that DRMGuard can consistently defend against various backdoor attacks. We also generalize four state-of-the-art defenses designed for classifiers to regression models, and compare DRMGuard with them. The results show that DRMGuard significantly outperforms all those defenses. The code will be open-sourced upon paper acceptance.

## 1 Introduction

Regression techniques are widely used to solve tasks where the goal is to predict continuous values. Unsurprisingly, similar to their classification counterparts, regression techniques have been revolutionized with deep learning and have achieved the state-of-the-art result in many real-world applications. Examples such as gaze estimation (Zhang et al., 2017b; 2019), head pose estimation (Borghi et al., 2017; Kuhnke & Ostermann, 2019), and facial landmark detection (Sun et al., 2013; Wu & Ji, 2019), among many others (Lathuilière et al., 2019). Unfortunately, deep regression models (DRM) inherited the vulnerabilities of deep neural networks (Gu et al., 2019; Liu et al., 2018b; Nguyen & Tran, 2021; 2020; Turner et al., 2019) and did not escape from the threat of backdoor attacks. Existing work (Sun et al., 2022) shows that an attacker can inject a backdoor trigger into a DRM such that it outputs an attacker-chosen target vector for any input stamped with an attacker-chosen backdoor trigger, while its predictions for clean inputs are unaffected. Therefore, given the wide adoption of DRM in many safety-critical applications such as driver attention monitoring (Abuelsamid, 2020; Berman, 2020), navigation of autonomous vehicles (Zeisl et al., 2015), and pedestrian attention monitoring (Raza et al., 2018; Schulz & Stiefelhagen, 2012), backdoor attacks raise severe safety concerns about the trustworthiness and robustness of DRMs.

Existing solutions to defend deep classification model (DCM) against backdoor attacks can be divided into *data-level* (Chen et al., 2018; Guo et al., 2023) and *model-level defenses* (Liu et al., 2019; Wang et al., 2019; 2022b). Data-level defenses aim to detect backdoored training or testing data, while model-level defenses aim to detect a potentially backdoored model and unlearn the backdoor behaviors. As we will discuss in Section 2, our work focuses on model-level defenses, as they are more realistic and do not assume the defender has access to the backdoored training or testing data.

However, existing model-level defenses (Wang et al., 2022b; 2019; Wu & Wang, 2021) are designed for DCMs. Our experiments show that they are ineffective when generalized and applied to DRMs

that are considered in this work. There are two underlying causes. First, distinct from DCMs (Wang et al., 2017) for which the output space is discretized into a few class labels, the output space of DRMs is continuous. Thus, it is infeasible (if not impossible) to enumerate and analyze all the potential target vectors using existing defenses designed for DCMs to determine the infected target (Wang et al., 2019) or the compromised neurons (Liu et al., 2019). Second, different from DCMs that adopt $\arg\max$ to obtain the final output, DRMs do not need $\arg\max$. This makes the backdoor behavior of the backdoored DRMs different from that of the backdoored DCMs. Specifically, for a backdoored DCM, the backdoor behavior is often triggered by the activation values of several neurons in the feature space (Wang et al., 2022b), whereas for a backdoored DRM, it is triggered by the activation values of all the neurons, which makes it harder to be detected or mitigated.

**Our work:** In this paper, we propose DRMGuard, the first framework to detect backdoored DRMs in the image domain. DRMGuard is applied to a DRM to reverse engineer a potential trigger function, based on which we make the decision on whether the model has been injected a backdoor or not. A major challenge to reverse engineering of the potential trigger function in the regression domain is that the output is defined in the continuous space. To address this challenge, we formulate the reverse engineering as an optimization problem, which is based on both output-space and feature-space characteristics of the backdoored DRMs that are observed in this paper.

To demonstrate the effectiveness of DRMGuard, we consider two regression tasks, and conduct extensive experiments on four datasets for state-of-the-art backdoor attacks. Our experimental results suggest that DRMGuard is consistently effective in defending both input-independent attacks, e.g., BadNets (Gu et al., 2019), and input-aware attacks, e.g., Input-aware dynamic attack (Nguyen & Tran, 2020). Furthermore, we adapt four state-of-the-art backdoor defenses, i.e., Neural Cleanse (Wang et al., 2019), FeatureRE (Wang et al., 2022b), ANP (Wu & Wang, 2021), and Fine-pruning (Liu et al., 2018a), designed for classifiers to regression models and compare DRMGuard with them. The results demonstrate that DRMGuard outperforms all of them by a large margin.

## 2 BACKGROUND AND RELATED WORK

**Backdoor Attacks:** Many backdoor attacks (Chen et al., 2017; Gu et al., 2019; Liu et al., 2018b; Phan et al., 2022; Wang et al., 2022a;c; Yao et al., 2019; Zhao et al., 2022) have been proposed for deep neural networks. They showed that an attacker can inject a backdoor into a classifier and make it output an attacker-chosen target class for any input embedded with an attacker-chosen backdoor trigger. Depending on whether the attacker uses the same backdoor trigger for different testing inputs, we categorize existing attacks into *input-independent attacks* (Chen et al., 2017; Gu et al., 2019; Liu et al., 2018b; Turner et al., 2019; Yao et al., 2019) and *input-aware attacks* (Koffas et al., 2022; Li et al., 2021b; Nguyen & Tran, 2021; 2020; Salem et al., 2022). For instance, Gu et al. (2019) proposed an input-independent backdoor attack by using a fixed pattern, e.g., a white patch, as the backdoor trigger. Recently, researchers proposed to use input-aware techniques, such as the warping process (Nguyen & Tran, 2021) and generative models (Nguyen & Tran, 2020), to generate dynamic triggers varying from input to input. When extending those attacks to DRMs (Sun et al., 2022), an attacker can inject a backdoor and make the model output a fixed vector (called *target vector*) for any testing input with the backdoor trigger. Lastly, backdoor attacks were also studied for graph neural networks (Xi et al., 2021; Zhang et al., 2021) and natural language processing (Shen et al., 2021). They are out of the scope of this paper as we focus on attacks in the image domain.

**Existing Defenses:** We categorize existing defenses against backdoor attacks into *data-level defenses* (Doan et al., 2020; Gao et al., 2019; Ma et al., 2023) and *model-level defenses* (Liu et al., 2022; 2019; Wu & Wang, 2021; Xiang et al., 2022; Zeng et al., 2022; Zheng et al., 2022). Data-level defenses detect whether a training example or a testing input is backdoored or not. They usually have two major limitations: 1) training data detection defenses (Chen et al., 2018) are not applicable for a given model that is already backdoored; and 2) testing input detection defenses (Doan et al., 2020) need to inspect each testing input at the running time and incur extra computation cost, and thus are undesired for latency-critical applications, e.g., gaze estimation (Zhang et al., 2020). Therefore, we focus on model-level defense in this work.

Model-level defenses detect whether a given model is backdoored or not, and state-of-the-art methods (Guan et al., 2022; Qiao et al., 2019; Wang et al., 2019; 2022b; Xiang et al., 2022) are based

on trigger reverse engineering. Specifically, they view each class as a potential target class and reverse engineer a backdoor trigger for it. Given the reverse-engineered backdoor triggers, they use statistical techniques to determine whether the classification model is backdoored or not. However, existing solutions are mainly designed for classification tasks that have categorical output. As we will show in this paper, they cannot be applied to DRMs. We note that Li et al. (2021a) also studied backdoor defense for DRMs, but they only considered a specific attack designed in their paper where the inputs of the regression model are low-dimensional vectors, i.e., five dimensional vectors. By contrast, we consider DRMs in the high-dimensional image domain.

## 3 DESIGN OF DRMGUARD

### 3.1 THREAT MODEL

**Deep regression model:** A deep regression model (DRM) is a deep neural network that maps an input to a vector, i.e., $f : \mathcal{X} \mapsto \mathcal{Y}$, where $\mathcal{X} \subset \mathbb{R}^{N_w \times N_h \times N_c}$ represents the input space with width $N_w$, height $N_h$, and channel $N_c$; and $\mathcal{Y} \in \mathbb{R}^d$ represents the $d$-dimensional output space. Given a training dataset $\mathcal{D}_{tr}$ that contains a set of training examples, we define the following loss $\frac{1}{|\mathcal{D}_{tr}|} \sum_{(x,y)\in\mathcal{D}_{tr}} \ell(f(x), y)$, where $(x, y)$ is a training example in $\mathcal{D}_{tr}$ and $\ell$ is the loss function for the regression task (e.g., $\ell_2$ loss) to update the parameters of $f$.

**Backdoor attacks:** We consider existing backdoor attacks for classification models (Gu et al., 2019; Nguyen & Tran, 2021; 2020; Turner et al., 2019), and adapt them to DRMs. Specifically, given a training dataset $\mathcal{D}_{tr}$, an attacker can add backdoor triggers to the training samples in $\mathcal{D}_{tr}$, and change their ground-truth annotations to an attacker-chosen vector, $y_T \in \mathcal{Y}$, known as the *target vector*. The attacker can manipulate the training process. The backdoored DRM performs well on benign inputs, but outputs the target vector $y_T$ when the backdoor trigger is present in the input. Formally, we define the backdoor attack for DRMs as:

$$f(x) = y, \ f(\mathcal{A}(x)) = y_T, \tag{1}$$

where $f$ is the backdoored DRM, $x \in \mathcal{X}$ is the benign input; $y \in \mathcal{Y}$ is the benign ground-truth annotation; and $\mathcal{A}$ is the trigger function that constructs the poisoned input from the benign input.

**Evaluation metric for backdoor attacks:** Given a set of poisoned inputs, we define attack error (AE) as the average regression error calculated from the output vectors and the target vector over all the poisoned inputs, to evaluate the performance of backdoor attacks on DRMs. AE can be regarded as the counterpart to the attack success rate for backdoor attacks on classification models.

**Assumptions and goals of the defender:** The defense goal is to identify if a DRM has been backdoored or not. Following existing defenses for backdoor attacks (Liu et al., 2019; Wang et al., 2019; 2022b), we assume the defender can access the trained DRM and a small benign dataset $\mathcal{D}_{be}$ with correct annotations.

### 3.2 OVERVIEW OF DRMGUARD

We propose DRMGuard to identify if a DRM has been backdoored by reverse engineering the trigger function $\mathcal{A}$. Figure 1 shows the overview of DRMGuard. We use a generative model $G_\theta$ to model $\mathcal{A}$. This allows us to model the trigger function for both input-independent and input-aware attacks. For a given DRM $f$ under examination, we split it into two submodels. Specifi-

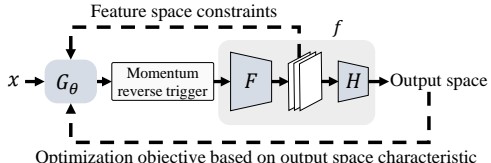

Figure 1: Overview of DRMGuard.

cally, we first use the submodel $F$ to map the original input $x$ to the feature space $F(x) \in \mathbb{R}^m$, i.e., the output space of the last but second layer of $f$. Then, we use the submodel $H$ to map the intermediate feature from the feature space to the final output space. This allows us to investigate the characteristics of the backdoored DRMs in both feature space and output space, based on which we formulate the optimization problem for reverse engineering. Moreover, we propose a strategy called *momentum reverse trigger* to reverse high-quality triggers.

### 3.3 Observations and Intuitions for Backdoored Deep Regression Model

Reverse engineering is performed by solving an optimization problem with constraints that are designed based on observations in the input (Wang et al., 2019) or the feature space (Wang et al., 2022b). Existing work (Wang et al., 2022b) shows that by using feature-space constraints, one can reverse both input-independent trigger (Gu et al., 2019) and input-aware trigger (Nguyen & Tran, 2021; 2020) for a backdoored deep classification model (DCM). Following this trend, we consider feature-space constraints when designing the reverse engineering for DRM. Below, we first discuss the difference in the feature space between backdoored DCM and backdoored DRM. Then, through theoretical analysis and experiments, we introduce the key observation for backdoored DRM.

**Difference between backdoored DCM and DRM:** A key observation for backdoored DCMs is that the backdoor behavior is represented by the activation values of several neurons in the feature space (Liu et al., 2019; Wang et al., 2022b). Specifically, when a trigger is present in the input, the activation values of the affected neurons will drop into a certain range, making the backdoored DCM output the attacker-chosen target class regardless of the activation values of the other neurons. This is because, after applying a series of operations to the feature vector, a backdoored DCM utilizes $\arg\max$ to obtain the final classification output. As long as the activation values of the affected neurons can make the target class have the highest probability, the influence of the other neurons on the final classification output will be eliminated by $\arg\max$. By contrast, the final regression output of a backdoored DRM is obtained by applying linear transformation (or followed by an activation function) to the feature vector without using $\arg\max$. Thus, the activation value of each neuron in the feature space contributes to the final output. This difference inspires us to take all the neurons into consideration when searching for the feature-space characteristics of backdoored DRMs, rather than looking at a few specific neurons only.

**Theoretical analysis and metrics:** We use $\{h_i\}_{i=1}^N$ and $\{h_i^p\}_{i=1}^N$ to denote the feature vectors extracted from a set of $N$ benign inputs $\{x_i\}_{i=1}^N$ and a set of poisoned inputs $\{\mathcal{A}(x_i)\}_{i=1}^N$, respectively, where $h_i = F(x_i) \in \mathbb{R}^m$ and $h_i^p = F(\mathcal{A}(x_i)) \in \mathbb{R}^m$. We use $y_{i,j}$ and $y_{i,j}^p$ to denote the $j$th component of the output vector $y_i = H(h_i) \in \mathbb{R}^d$ and $y_i^p = H(h_i^p) \in \mathbb{R}^d$. $y_{i,j}^p$ is calculated by:

$$y_{i,j}^p = \Omega(w_j \cdot h_i^p + b_j) = \Omega(\|w_j\|_2 \|h_i^p\|_2 \cos\alpha_{i,j}^p + b_j), \tag{2}$$

where $\Omega(\cdot)$ is the activation function; $w_j \in \mathbb{R}^m$ and $b_j \in R$ are the weights vector and the bias of $H$ for the $j$th component of the output vector, respectively; $\alpha_{i,j}^p$ is the angle between $h_i^p$ and $w_j$. Based on Equation 1, we have $y_{1,j}^p \approx y_{2,j}^p \approx \cdots \approx y_{N,j}^p$ if $f$ is backdoored, which means $\sigma^2(\{y_{i,j}^p\}_{i=1}^N)$ is a small positive value, where $\sigma^2(\cdot)$ is the variance function. As shown in Equation 2, the value of $\sigma^2(\{y_{i,j}^p\}_{i=1}^N)$ is influenced only by $\|h_i^p\|_2$ and $\alpha_{i,j}^p$, as $\|w_j\|_2$ and $b_j$ are constant for a given DRM. Moreover, when $f$ is backdoored, $\sum_{j=1}^d \sigma^2(\{y_{i,j}^p\}_{i=1}^N)/d$ is a small positive value and influenced by $\|h_i^p\|_2$ and $\alpha_i^p$, where $\alpha_i^p = \{\alpha_{i,1}^p, ..., \alpha_{i,d}^p\} \in \mathbb{R}^d$. We use $\alpha_{i,j}$ to denote the angle between $h_i$ and $w_j$, and define $\alpha_i$ as $\alpha_i = \{\alpha_{i,1}, ..., \alpha_{i,d}\} \in \mathbb{R}^d$.

To further investigate how $\|h_i^p\|_2$ and $\alpha_i^p$ influence $\sum_{j=1}^d \sigma^2(\{y_{i,j}^p\}_{i=1}^N)/d$, we introduce **the ratio of norm variance (RNV)** and **the ratio of angle variance (RAV)**, as two feature-space metrics:

$$\text{RNV} = \sigma^2(\{\|h_i^p\|_2\}_{i=1}^N)/\sigma^2(\{\|h_i\|_2\}_{i=1}^N) \text{ and } \text{RAV} = \frac{1}{d}\sum_{j=1}^d \sigma^2(\{\alpha_{i,j}^p\}_{i=1}^N)/\sigma^2(\{\alpha_{i,j}\}_{i=1}^N). \tag{3}$$

Specifically, RNV compares the dispersion of $\{\|h_i^p\|_2\}_{i=1}^N$ and $\{\|h_i\|_2\}_{i=1}^N$, while RAV compares the dispersion of $\{\alpha_i^p\}_{i=1}^N$ and $\{\alpha_i\}_{i=1}^N$. RNV $\ll 1$ indicates that when triggers are present in the inputs, the feature vectors extracted by $F$ have similar norms. RAV $\ll 1$ means that the variance of angles between $\{h_i^p\}_{i=1}^N$ and $w_j$ are much smaller than that between $\{h_i\}_{i=1}^N$ and $w_j$ for $j = 1, ..., d$.

**Observations:** We use four backdoor attacks, i.e., BadNets (Gu et al., 2019), Input-aware dynamic attack (IA) (Nguyen & Tran, 2020), WaNet (Nguyen & Tran, 2021), and Clean Label (Turner et al., 2019), to train backdoored DRMs on MPIIFaceGaze dataset (Zhang et al., 2017a) and Biwi Kinect dataset (Fanelli et al., 2013). Table 1 shows the RNV and the RAV of the backdoored DRMs that are trained by different backdoor attacks on the two datasets. The key observation is that RAV is significantly smaller than 0.1 in all the examined scenarios. To further explore this observation, the scatter plots in Figure 2 visualize $\{\alpha_i^p\}_{i=1}^N$ and $\{\alpha_i\}_{i=1}^N$ in all the examined cases. We can see

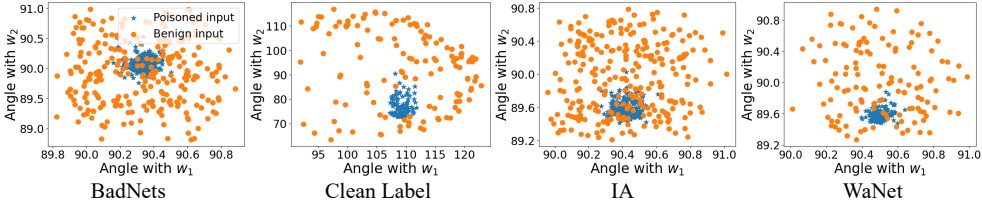

(a) Backdoored DRMs trained on MPIIFaceGaze under different backdoor attacks ($d = 2$).

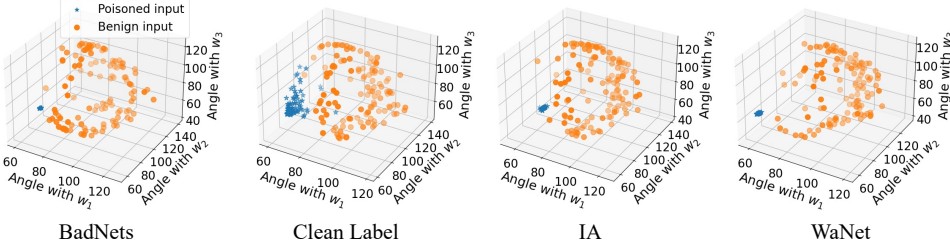

(b) Backdoored DRMs trained on Biwi Kinect under different backdoor attacks ($d = 3$).

Figure 2: The plots of $\{\alpha_i^p\}_{i=1}^N$ and $\{\alpha_i\}_{i=1}^N$ (in degree) for backdoored DRMs trained on (a) MPI-IFaceGaze dataset and (b) Biwi Kinect dataset. The spread of the data points shows that the angles of the poisoned inputs are highly concentrated, while the angles of the benign inputs are scattered.

that the angles of the poisoned inputs are highly concentrated, while the angles of the benign inputs are scattered, meaning that $\sigma^2(\{\alpha_{i,j}^p\}_{i=1}^N) \ll \sigma^2(\{\alpha_{i,j}\}_{i=1}^N)$ for $j = 1, \cdots, d$. We summarize this observation in the feature space as follows.

**Key observation for backdoored DRMs in the feature space:** *Consider a set of benign inputs* $\{x_i\}_{i=1}^N$ *and a set of poisoned inputs* $\{\mathcal{A}(x_i)\}_{i=1}^N$. *Let* $W = \{w_1, w_2, ..., w_d\}^\mathsf{T} \in \mathbb{R}^{m \times d}$ *be the weights matrix of* $H$, *where* $w_j \in \mathbb{R}^m$. *Then, we have the following observation:*

$$\sigma^2\left(\{\mathcal{B}(F(\mathcal{A}(x_i)), w_j)\}_{i=1}^N\right) \ll \sigma^2\left(\{\mathcal{B}(F(x_i), w_j)\}_{i=1}^N\right) \ \text{for} \ j = 1, 2, ..., d, \tag{4}$$

*where* $\mathcal{B}(v_1, v_2) = \arccos(v_1 \cdot v_2)/(\|v_1\|_2 \|v_2\|_2)$; *and* $\mathcal{A}$ *is the trigger function.*

Lastly, Table 1 shows that RNV is close to or greater than 0.1 for backdoored DRMs trained by BadNets, IA, and WaNet on MPIIFaceGaze, while for the other examined cases, RNV is significantly smaller than 0.1. This is because, as shown in Figure 2, when RNV $\geq 0.1$, each component of $\alpha_i^p$, i.e., $\alpha_{i,j}^p$ for $j = 1, \cdots, d$, is almost 90°, for $i = 1, \cdots, N$, meaning that $\cos \alpha_{i,j}^p$ is almost zero. Thus, $y_{i,j}^p$ is insensitive to the change of $\|h_i^p\|_2$,

Table 1: The RNV and RAV for four attacks on two datasets. In all the examined cases, RAV is significantly smaller than 0.1.

| Attack | MPIIFaceGaze | | Biwi Kinect | |
|---|---|---|---|---|
| | RAV | RNV | RAV | RNV |
| BadNets | 0.0433 | 1.4499 | 0.0002 | 0.0012 |
| IA | 0.0489 | 2.5714 | 0.0015 | 0.0046 |
| Clean Label | 0.0328 | 0.0428 | 0.0803 | 0.0341 |
| WaNet | 0.0311 | 0.8528 | 0.0003 | 0.0015 |

which allows RNV to be similar to or even larger than 0.1 but still maintain a low $\sigma^2(\{y_{i,j}^p\}_{i=1}^N)$.

### 3.4 METHODOLOGY

**Reverse engineering for DRM:** One major challenge for reverse engineering for DRM is that the target vector $y_T$ is defined in the continuous output space. As a result, it is impossible to enumerate and analyze all the potential target vectors using existing reverse engineering methods designed for DCM, which treat each class as a potential target class and reverse the trigger for it (Wang et al., 2019; 2022b). To resolve this challenge, we propose to reverse engineer $\mathcal{A}$ by minimizing: $\sum_{j=1}^d \sigma^2\left(\{f_j(G_\theta(x_i))\}_{i=1}^N\right)/d$, where $f_j(G_\theta(x_i))$ is the $j$th component of $f(G_\theta(x_i)) \in \mathbb{R}^d$. This is intuitive, as for a backdoored $f$ the term $\frac{1}{d}\sum_{j=1}^d \sigma^2\left(\{y_{i,j}^p\}_{i=1}^N\right)$ will be a small positive value. Thus, we can search for the target vector in the continuous output space by learning a trigger function $\mathcal{A}$, modeled by $G_\theta$, that can mislead $f$ to map different inputs to the target vector without enumerating all the potential target vectors. Moreover, based on the key observation of backdoored DRMs defined in Inequation 4, we introduce the feature-space regularization term in the optimiza-

tion. Formally, we define the optimization problem for the reverse engineering for DRM as:

$$\theta^* = \min_\theta \frac{\lambda_1}{d} \sum_{j=1}^d \sigma^2 \left( \{f_j\left(x_i'\right)\}_{i=1}^N \right) + \frac{1}{N} \sum_{i=1}^N \|x_i' - x_i\|_1 + \lambda_2 r_f, \tag{5}$$

where $r_f = \frac{1}{d} \sum_{j=1}^d \sigma^2(\{\mathcal{B}(F(x_i'), w_j)\}_{i=1}^N)/\sigma^2(\{\mathcal{B}(F(x_i), w_j)\}_{i=1}^N)$; $x_i' = G_\theta(x_i)$; and $\lambda_1$ and $\lambda_2$ are the weights for the first and third objectives, respectively. The first objective in the optimization problem aims to reverse engineer the poisoned inputs $\{G_\theta(x_i)\}_{i=1}^N$ that lead to the same output vector, regardless of their actual contents. The second objective is the input-space regularization term (Wang et al., 2019; 2022b) that ensures the transformed input $G_\theta(x_i)$ is similar to the benign input $x_i$. The final objective is the feature-space regularization term, which can lead to a much lower variance for $\{\hat{\alpha}_{i,j}^p\}_{i=1}^N$ than that for $\{a_{i,j}\}_{i=1}^N$, where $\hat{\alpha}_{i,j}^p$ is the angle between $F(G_\theta(x_i))$ and $w_j$.

**Momentum reverse trigger:** Directly solving the optimization problem defined in Equation 5 can result in a sub-optimal solution (illustrated in Appendix A.1): the algorithm focuses on the important regions in the input image, to which the DRM pay more attention, and directly adds perturbations to these regions to *destroy* the task-related features. To avoid such a trivial solution, we propose the *momentum reverse trigger* to assign different weights to different regions to balance the attention of the DRM on the image, such that the algorithm can pay attention to all the image pixels and search for the trigger that are injected into both important and unimportant regions. Details about the momentum reverse trigger are given in Appendix A.1. After introducing the momentum reverse trigger into Equation 5, we use $\mathcal{OPT}$-DRMGuard to denote the final optimization problem.

**Backdoor identification:** By solving the optimization problem $\mathcal{OPT}$-DRMGuard, we can obtain the perturbation $\|x_i' - x_i\|$ that transforms input $x_i$ to the potential target vector. We observe that the amount of perturbation required to transform the input to the potential target vector for a backdoored DRM is significantly smaller than that for a benign DRM. Based on this observation, DRMGuard introduces the metric $\mathcal{I}(f) = \mathbb{1}(\frac{1}{N} \sum_{i=1}^N \|x_i' - x_i\|_1, \epsilon\|\hat{x}\|_1)$, to identify if a given deep regression model is backdoored or not, where $\|\hat{x}\|_1$ is the input image that has the maximum $L1$ norm in the benign dataset $\mathcal{D}_{be}$; $\epsilon$ is a constant; and $\mathbb{1}$ is the indicator function that returns 1 (backdoored DRM) if $\frac{1}{N} \sum_{i=1}^N \|x_i' - x_i\|_1 < \epsilon\|\hat{x}\|_1$ and 0 (benign DRM) otherwise. We set $\epsilon = 0.03$.

**Backdoor mitigation:** Once a given DRM is identified as a backdoored DRM, DRMGuard uses the reversed trigger function $G_\theta$ and the available benign dataset $\mathcal{D}_{be}$ to generate a reversed poisoned dataset $\mathcal{D}_{rp}$ with the original correct annotations. Then, DRMGuard fine-tunes the given backdoored DRM by using $\mathcal{D}_{be}$ and $\mathcal{D}_{rp}$ to unlearn the backdoor behavior.

## 4 EVALUATION

### 4.1 EXPERIMENTAL SETUP

**Regression tasks:** We consider two regression tasks, i.e., gaze estimation and head pose estimation. Gaze estimation tracks where the subject is looking at, and plays a key role in a series of safety-critical applications, such as user authentication (Eberz et al., 2019; Katsini et al., 2020) and driver distraction detection (Berman, 2020). Similarly, head pose estimation has also been used in many safety-related applications, such as the driver assistance system (Jha & Busso, 2016; Murphy-Chutorian et al., 2007) and pedestrian attention monitoring (Schulz & Stiefelhagen, 2012).

**Datasets:** We consider four benchmark datasets, i.e., MPIIFaceGaze (Zhang et al., 2019), ColumbiaGaze (Smith et al., 2013), Biwi Kinect (Fanelli et al., 2013), and Pandora (Borghi et al., 2017). For each dataset, we randomly select 80% and 10% of the images from the dataset to form the training dataset $\mathcal{D}_{tr}$ and the benign dataset $\mathcal{D}_{be}$, respectively. We use the remainder as the testing set $\mathcal{D}_{te}$ to evaluate the performance of backdoor mitigation. Details of datasets can be found in Appendix A.2.

**Backdoor attacks:** We consider four state-of-the-art backdoor attacks, including two input-independent attacks, i.e., BadNets (Gu et al., 2019) and Clean Label (Turner et al., 2019), and two input-aware attacks, i.e., Iuput-aware dynamic attack (IA) (Nguyen & Tran, 2020) and WaNet (Nguyen & Tran, 2021). We detail how to adapt these backdoor attacks to DRMs and the effectiveness of them on DRMs in Appendix A.3.

Table 2: Backdoor identification performance of DRMGuard on MPIIFaceGaze for different attacks. DRMGuard can defend various attacks.

| Attack | TP | FP | FN | TN | Acc |
|---|---|---|---|---|---|
| BadNets | 10 | 1 | 0 | 9 | 95% |
| IA | 10 | 1 | 0 | 9 | 95% |
| Clean Label | 10 | 1 | 0 | 9 | 95% |
| WaNet | 10 | 1 | 0 | 9 | 95% |

Table 3: Backdoor identification performance of DRMGuard on different datasets for WaNet. DRMGuard is effective on various datasets.

| Dataset | TP | FP | FN | TN | Acc |
|---|---|---|---|---|---|
| MPIIFaceGaze | 10 | 1 | 0 | 9 | 95% |
| ColumbiaGaze | 8 | 4 | 2 | 6 | 70% |
| Biwi Kinect | 10 | 0 | 0 | 10 | 100% |
| Pandora | 7 | 0 | 3 | 10 | 85% |

**Compared defenses:** For backdoor identification, we compare DRMGuard with two state-of-the-art methods, i.e., Neural Cleanse (NC) (Wang et al., 2019) and FeatureRE (Wang et al., 2022b). Since the output space of DRM is continuous, we generalize them from classification models to regression models by taking the potential target vector $y_t$ as the optimization variable. We provide details of the generalizations in Appendix A.4. For backdoor mitigation, we compare DRMGuard with two state-of-the-art methods, i.e., ANP (Wu & Wang, 2021) and Fine-pruning (Liu et al., 2018a).

**Evaluation metrics:** Following existing work (Wang et al., 2022b), we use the identification accuracy as the performance metric. In detail, given a set of DRMs including benign and backdoored DRMs, the identification accuracy is defined as the percentage of correctly classified DRMs over all examined DRMs. We also report the number of True Positives (TP), i.e., correctly identified backdoored DRMs, False Positives (FP), i.e., benign DRMs recognized as backdoored DRMs, False Negatives (FN), i.e., backdoored DRMs identified as benign DRMs, and True Negatives (TN), i.e., correctly recognized benign DRMs. Moreover, we use ROC-AUC score to compare backdoor identification performance between DRMGuard, NC, and FeatureRE, after obtaining the average perturbations on $\mathcal{D}_{be}$ for benign DRMs and backdoored DRMs.

To evaluate the performance of DRMGuard on backdoor mitigation, we generate a poisoned dataset $\mathcal{PD}_{te}$ by applying trigger function to all the images in $\mathcal{D}_{te}$. We define defending attack error (DAE) as the average regression error calculated from the output vectors and the correct annotations over all the images in $\mathcal{PD}_{te}$. Details of regression errors for the two examined regression tasks are given in Appendix A.2. We use DAE and AE on $\mathcal{PD}_{te}$ as the evaluation metrics for backdoor mitigation.

**Defense settings:** Unless otherwise mentioned, we set $\lambda_1 = 20$ and $\lambda_2 = 800$ for gaze estimation, and set $\lambda_1 = 10$ and $\lambda_2 = 100$ for head pose estimation, given task difference. We use ResNet18 (He et al., 2016) (without the dense layer) to implement $F$, and a dense layer without activation function to implement $H$. We consider gaze estimation task with MPIIFaceGaze dataset and the state-of-the-art input-aware attack WaNet.

## 4.2 EVALUATION RESULTS ON BACKDOOR IDENTIFICATION

**DRMGuard is effective for backdoor identification:** We conduct three experiments to evaluate the backdoor identification performance. First, we evaluate the performance of DRMGuard in identifying backdoored DRMs trained by different attacks. Specifically, for each of the four backdoor attacks, i.e., BadNets, Clean Label, WaNet, and IA, we train ten benign DRMs and ten backdoored DRMs on MPIIFaceGaze dataset. The results are shown in Table 2, which indicate that DRMGuard can identify backdoored DRMs trained by both input-independent and input-aware attacks, at an average accuracy of 95%. Moreover, we visualize the estimation of the target vector during the training process and the reversed trigger in the Appendix A.5 and A.6, respectively.

Second, we examine the backdoor identification capability of DRMGuard on different regression tasks and datasets, i.e., MPIIFaceGaze, ColumbiaGaze, Biwi Kinect, and Pandora. Specifically, we train ten benign DRMs and ten backdoored DRMs using WaNet for each dataset. The results are shown in Table 3. The average identification accuracy of DRMGuard on different datasets is 87.5%, which demonstrates the effectiveness of DRMGuard on various regression tasks and datasets.

Finally, we consider the scenario where the DRM is backdoored by multiple trigger functions with different target vectors. We report the attacking details and evaluation results in Appendix A.7. In brief, the results show that our method is effective on identifying DRMs with multiple backdoors.

**DRMGuard outperforms state-of-the-art defenses:** Table 4 shows the ROC-AUC scores of DRMGuard, NC, and FeatureRE for four backdoor attacks. We also report the scores when ap-

Table 4: ROC-AUC scores of different methods on MPIIFaceGaze for different attacks. DRMGuard significantly outperforms NC and FeatureRE.

| Attack | NC | FeatureRE | DRMGuard |
|---|---|---|---|
| BadNets | 0.270 | 0.730 | **1.000** |
| IA | 0.300 | 0.560 | **1.000** |
| WaNet | 0.940 | 0.560 | **1.000** |
| Clean Label | 0.005 | 0.545 | **1.000** |
| All attacks | 0.379 | 0.599 | **1.000** |

Table 5: Performance of backdoor mitigation for different attacks. DRMGuard can mitigate backdoor behaviors for various attacks.

| Attack | Undefended | | DRMGuard | |
|---|---|---|---|---|
| | AE | DAE | AE | DAE |
| BadNets | 3.25 | 14.85 | 17.21 | 3.59 |
| IA | 3.19 | 14.40 | 15.69 | 3.50 |
| Clean Label | 0.72 | 15.43 | 16.42 | 2.51 |
| WaNet | 1.31 | 15.90 | 15.36 | 3.29 |

plying the four backdoor attacks simultaneously. As shown, the ROC-AUC score of DRMGuard is 1.000 in all the examined cases, which is significantly higher than that of NC and FeatureRE. Besides, we notice that FeatureRE fails to find a trigger function that enables the backdoored DRM to map different inputs to similar output vectors, which confirms our analysis that the feature-space characteristic for backdoored DCM (Wang et al., 2022b) does not hold for backdoored DRM.

## 4.3 EVALUATION RESULTS ON BACKDOOR MITIGATION

We train backdoored DRMs using BadNets, Clean Label, IA, and WaNet on MPIIFaceGaze. Table 5 shows AE and DAE of the undefended and mitigated backdoored DRMs, which indicate that DRMGuard can mitigate backdoor behaviors for various attacks. Specifically, DRMGuard can significantly increase AE and decrease DAE for all the attacks, which indicates that the output vectors of DRMs are far away from the target vector and close to the correct annotations after backdoor mitigation, even though triggers are injected in the inputs.

Table 6: Performance of different methods on backdoor mitigation. DRMGuard outperforms baselines.

| Method | AE | DAE |
|---|---|---|
| Undefended | 1.31 | 15.90 |
| DRMGuard | **15.36** | **3.29** |
| Fine-tuning | 13.96 | 4.32 |
| Fine-pruning | 6.68 | 16.82 |
| ANP | 4.92 | 13.13 |

We compare DRMGuard with ANP and Fine-pruning on backdoor mitigation. We also consider a baseline, i.e., Fine-tuning, which directly uses the benign dataset $\mathcal{D}_{be}$ to fine tune the backdoored DRM. We report AE and DAE for different methods after backdoor mitigation in Table 6. The AE for DRMGuard is significantly larger than that for other methods, while the DAE for DRMGuard is much smaller than that for other methods, which shows the superiority of DRMGuard on backdoor mitigation. Moreover, Fine-pruning and ANP are built upon the feature-space characteristics of backdoored DCM and perform terribly on backdoored DRMs. This also confirms our analysis that the feature-space characteristics of backdoored DRMs are different with that of backdoored DCM.

## 4.4 ABLATION STUDIES

**Impact of weights and the size of benign dataset:** To investigate the impact of $\lambda_1$ and $\lambda_2$ in Equation 5 on the performance of backdoor identification, we vary $\lambda_1$ and $\lambda_2$ from 10 to 30 and from 600 to 800, respectively. Moreover, we study the impact of the size of $\mathcal{D}_{be}$ on the identification performance by changing the ratio $p$ of benign dataset to the original whole dataset from 5% to 15%.

We report the results in Table 7. We observe that the performance of DRMGuard is insensitive to $\lambda_1$, as the identification accuracy is almost stable with different $\lambda_1$. However, DRMGuard is sensitive to $\lambda_2$ and the identification accuracy increases with $\lambda_2$. This observation proves that the proposed feature-space regularization term is important for the identification of backdoored DRMs. We also observe that as $p$ decreases from 15% to 5%, the identification accuracy and the number of TN decrease, while the number of TP remains stable. This is because, compared to a large $p$, it is easier to find a small amount of perturbation that can lead to the backdoor behavior on a small $p$ for benign models. However, the identification accuracy is still 90% even when $p = 5\%$.

**Impact of feature-space regularization term (FSRT):** We remove FSRT from $\mathcal{OPT}$-DRMGuard and show the results in Table 8, which indicate that all the DRMs are classified as backdoored DRMs. We further observe that without the FSRT, DRMGuard cannot find a trigger function that can map different inputs to similar output vectors. As a result, DRMGuard solves the optimization problem by focusing on minimizing the distance between the poisoned and benign images, and returns a small amount of perturbations, which leads to the misclassification of backdoored DRMs.

Table 7: Ablation study on the impact of different value of $\lambda_1$, $\lambda_2$ and $p$.

| Metric | Different $\lambda_1$ | | | Different $\lambda_2$ | | | Different $p$ | | |
|--------|------|------|------|------|------|------|------|------|------|
| | 10 | 20 | 30 | 600 | 800 | 1000 | 5% | 10% | 15% |
| TP | 10 | 10 | 10 | 10 | 10 | 10 | 10 | 10 | 10 |
| FP | 0 | 1 | 1 | 5 | 1 | 0 | 2 | 1 | 0 |
| FN | 0 | 0 | 0 | 0 | 0 | 0 | 0 | 0 | 0 |
| TN | 10 | 9 | 9 | 5 | 9 | 10 | 8 | 9 | 10 |
| Acc | 100% | 95% | 95% | 75% | 95% | 100% | 90% | 95% | 100% |

Table 8: Ablation study on FSRT and MTR.

| Method | TP | FP | FN | TN | Acc |
|--------|----|----|----|----|-----|
| w/o FSRT | 10 | 10 | 0 | 0 | 50% |
| w/o MTR | 10 | 10 | 0 | 0 | 50% |

Table 9: Evaluation results on adaptive attack.

| Attack | AE | DAE | Acc |
|--------|------|-------|-----|
| WaNet | 1.51 | 15.99 | 95% |
| Adaptive | 5.71 | 15.01 | 95% |

**Impact of momentum reverse trigger (MTR):** We remove the MTR from $\mathcal{OPT}$-DRMGuard and report the identification results in Table 8. As shown, all the benign DRMs are classified as backdoored DRMs. This is because, without the MTR, DRMGuard will find a small amount of perturbation and add it to the eye regions to destroy gaze-related features and fool the benign models. In this way, the poisoned images reversed from different images are transformed by $f$ to similar output vectors, and DRMGuard fails to correctly recognize benign DRMs.

## 4.5 ADAPTIVE ATTACKS

When the attacker has the full knowledge of DRMGuard, one potential adaptive attack that can bypass our method is to force the left and the right terms in Inequation 4 to have similar values. Based on this intuition, we design an adaptive attack that adds an additional loss term $L_{adp}$ with a weight $\lambda_{adp}$ to the original loss function of the chosen backdoor attack. We define $L_{adp}$ as:

$$L_{adp} = \left| 1 - \frac{1}{d} \sum_{j=1}^{d} \sigma^2 \left( \left\{ \mathcal{B}(F(\mathcal{A}(x_i)), w_j) \right\}_{i=1}^{N_p} \right) / \sigma^2 \left( \left\{ \mathcal{B}(F((x_i), w_j) \right\}_{i=1}^{N_b} \right) \right|, \tag{6}$$

where $N_p$ and $N_b$ are the numbers of poisoned inputs and benign inputs in a minibatch. The loss term $L_{adp}$ tries to break the feature-space observation by enforcing RAV to be close to one. We generate ten backdoored DRMs by the adaptive attack with $\lambda_{adp} = 0.02$. Table 9 shows the identification accuracy and the averaged AE and DAE over ten backdoored DRMs. The AE of the adaptive attack is significantly higher than that of WaNet. This proves that our feature-space observation of the backdoored DRM is the key characteristic leading to the backdoor behavior. The adaptive attack cannot reduce the identification accuracy of our method.

## 5 DISCUSSION AND LIMITATION

**Discussion.** To further investigate the performance of DRMGuard, we evaluate DRMGuard by considering more backdoor attacks, different architectures of DRMs, and a larger set of DRMs in Appendix A.8. The results show DRMGuard can consistently defend against various backdoor attacks and can be generalized to different architectures. Also, DRMGuard maintains a similar identification accuracy on a larger set of DRMs that contains more backdoored and benign DRMs.

**Limitation.** Similar to backdoor defenses (Wang et al., 2019; 2022b) for DCM, our method requires a small benign dataset to identify backdoored DRM and mitigate backdoor behaviors.

## 6 CONCLUSION

We propose the first backdoor identification method DRMGuard for deep regression models in the image domain. Our method fills in the gap where existing backdoor identification methods only focus on deep classification models. Our comprehensive evaluation shows that our method can defend against both input-independent and input-aware backdoor attacks on various datasets.

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
