## A   APPENDIX

**Roadmap:**   We provide the details of the momentum reverse trigger in subsection A.1. The details of two regression tasks and four datasets are discussed in subsection A.2. Subsection A.3 describes the details of backdoor attacks and discusses how to generalize them to regression tasks. Then, we show the details about how to generalize Neural Cleanse (Wang et al., 2019) and FeatureRE (Wang et al., 2022b) to the deep regression model (DRM) in subsection A.4. Next, we visualize the estimation of the target vector during the reverse engineering process and the reversed poisoned images in subsection A.5 and subsection A.6, respectively. After that, we detail multi-target backdoor attack for DRMs in subsection A.7. We evaluate DRMGuard on more backdoor attacks, different architectures of DRMs, and a larger set of DRMs in Appendix A.8. We show that our idea of minimizing the variance can be extended to classification domain for trigger reverse engineering in Appendix A.9. Finally, we report the technical details of DRMGuard in subsection A.10.

**Experiment environment:**   We conduct experiments with Python 3.7.13 and Tensorflow 2.9.0 on an Ubuntu 20.04 machine with a NVIDIA A10 GPU.

### A.1   DETAILS OF MOMENTUM REVERSE TRIGGER

**Illustration of the sub-optimal solution:**   We illustrate the sub-optimal solution of the optimization problem defined in Equation 5 in Figure 3. Specifically, we use BadNets (Gu et al., 2019) to train a backdoored DRM on MPIIFaceGaze dataset for gaze estimation, where the trigger is a red square, added at the right bottom corner of the input images, as shown in Figure 3(b). Then, we optimize $G_\theta$ by solving the optimization problem defined in Equation 5. We show the residual map between the benign image and the reversed poisoned image in Figure 3(d). We can see that solving the optimization problem 5 fails to reverse the trigger but adds perturbations to the eyes region that contains the most important features for gaze estimation (Zhang et al., 2017a).

**Technical details:**   The idea of momentum reverse trigger is to assign different weights to different regions to balance the attention of the DRM on the image, such that the algorithm can pay attention to all the pixels and search for the trigger that are injected into both important and unimportant regions. To do this, we first generate an *attention map* $\mathcal{T}(x_i) \in \mathbb{R}^{N_w \times N_h}$ for each input $x_i$ based on the gradient of $f$ w.r.t. $x_i$. In detail, for each pixel $x_i[a, b]$, we obtain the corresponding value $\mathcal{T}(x_i)[a, b]$ in the attention map by $\mathcal{T}(x_i)[a, b] = \sum_{c=1}^{N_c} |\partial f / \partial x_i[a, b, c]|$. Then, we re-scale the attention map to $[0, 1)$ by dividing each component of the attention map by a number that is larger than the maximum value in the attention map. Then, instead of directly feeding $G_\theta(x_i)$ to $f$, we use the following image as the poisoned image and feed it to the DRM:

$$x_i' = G_\theta(x_i) \odot (1 - \mathcal{T}(x_i)) + x_i \odot \mathcal{T}(x_i). \tag{7}$$

### A.2   DETAILS OF REGRESSION TASKS AND DATASETS

#### A.2.1   DETAILS OF REGRESSION TASKS

We consider two regression tasks, i.e., gaze estimation and head pose estimation. Below, we introduce the details of each regression task.

**Gaze estimation:**   We consider 3D full-face gaze estimation, where the gaze estimation model aims to estimate the 3D gaze direction from the facial image. The 3D gaze direction is represented by a two-dimensional vector, denoting the yaw and pitch angles of the gaze direction. Following the existing works on gaze estimation (Zhang et al., 2020; 2017a), we use angular error as the regression error, which is defined as the angle between the estimated and the real gaze directions.

**Head pose estimation:**   We consider a head pose estimation model that takes a monocular image input and outputs a three-dimensional vector to denote the Eular angle (yaw, pitch, roll) of head pose (Gupta et al., 2019). We use $L1$ loss as the regression error.

#### A.2.2   DETAILS OF DATASETS

We consider four benchmark datasets in our evaluation and introduce the details of these datasets in the following. We also introduce the training details of benign DRMs on each dataset.

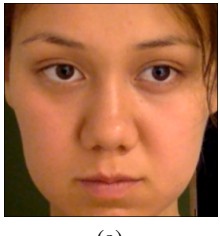 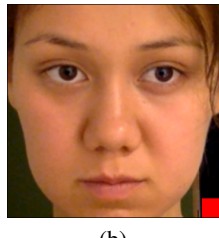 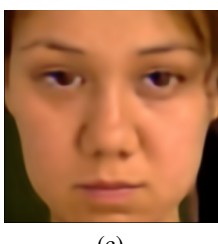 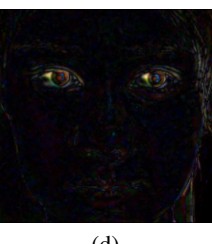

(a)  (b)  (c)  (d)

Figure 3: The visualization of the sub-optimal solution: (a) the benign image; (b) the poisoned image; (c) the reversed poisoned image when directly solving the optimization problem defined in Equation 5; and (d) the residual map between the benign and reversed poisoned images. Solving the optimization problem 5 fails to reverse the trigger but adds perturbations to the image region that contains the most important features for gaze estimation.

**MPIIFaceGaze** (Zhang et al., 2019) is a benchmark dataset for gaze estimation and is collected from 15 subjects in their daily life. Each subject contains 3,000 images under different backgrounds, illumination conditions, and head poses. The image resolution is $224 \times 224$. We use Adam optimizer with a learning rate of $0.0001$ to train the model for 10,000 training steps.

**ColumbiaGaze** (Smith et al., 2013) is a gaze estimation dataset collected from 56 subjects. Each subject has 105 facial images. We crop the facial patches from the original images and resize them to $224 \times 224$. We use Adam optimizer with an initial learning rate of $0.001$ to train the model for 10,000 training steps. The learning rate is multiplied by 0.1 at 5,000 and 9,000 training steps.

**Biwi Kinect Dataset** (Fanelli et al., 2013) is used for head pose estimation. It is collected from 24 subjects, and each subject has 400 to 900 images. Note that we use the *cropped faces of Biwi dataset (RGB images)* released by (Borghi et al., 2017). We resize the image to $112 \times 112$. We use Adam optimizer with a learning rate of $0.0001$ to train the model for 5,000 training steps.

**Pandora Dataset** (Borghi et al., 2017) is a dataset for head pose estimation, which includes more than 250k RGB and depth images. We use the pre-processed dataset, *Cropped faces (RGB images)*, released by the authors, where the facial images are cropped from the original images. The pre-processed dataset has 100 subjects, and contains more than 120,000 images. We use Adam optimizer with an initial learning rate of $0.0001$ to train the model for 10,000 training steps. The learning rate is multiplied by 0.1 at 5,000 training steps.

### A.3    DETAILS OF BACKDOOR ATTACKS

In this subsection, we describe the details of the backdoor attacks used in our evaluation and how to adapt them to DRMs. We also show the effectiveness of backdoor attacks on DRMs.

**BadNets** (Gu et al., 2019) uses a fixed pattern as the backdoor trigger, and the poisoned inputs are generated by pasting the backdoor trigger on the inputs. In our experiments, we use a $20 \times 20$ red patch located at the right-bottom corner as the backdoor trigger. The poisoning rate is $5\%$.

**Clean Label** (Turner et al., 2019) also uses a fixed pattern as the backdoor trigger. To be more stealthy, the trigger is only applied to images that belong to the target class for classification tasks. In our experiments, we use a $20 \times 20$ red patch located at the right-bottom corner as the backdoor trigger. To generalize Clean Label to regression tasks, we apply the trigger to the inputs whose annotations are "close" to the target vector. Formally, we consider the images whose annotations $y$ satisfy $\|y - y_T\| \leq \delta$ as the target group to be poisoned. We apply the PGD attack to half of the images in the target group to generate adversarial samples. To improve the performance of backdoor attacks, we apply the trigger to both the adversarial samples and another half of the images in the target group. This is because if we only apply trigger to the adversarial samples, the trained DRM will regard the perturbations generated by the PGD attack as the backdoor trigger and ignore the attacker-defined trigger pattern. Annotations of poisoned images are changed to the target vector.

**WaNet** (Nguyen & Tran, 2021) generates stealth backdoor triggers through image warping techniques. The trigger is inserted into the images by applying the elastic warping operation. Note that,

WaNet needs to modify the standard training process to train the backdoored DRM, while BadNets and Clean Label follow the standard way to train the backdoored DRM. To improve the performance of the attack, we set the warping strength to 1. We set the grid sizes for gaze estimation datasets and head pose estimation datasets to 28 and 14, respectively.

**Input-aware dynamic attack (IA)** (Nguyen & Tran, 2020) generates dynamic backdoor triggers by using a trainable trigger generator, which takes benign images as inputs and outputs triggers varying from input to input. A trigger generated by an input image cannot be used on another one. Similar to WaNet, IA also needs to modify the training process.

Table 10: Effectiveness of different backdoor attacks on DRMs.

| Metric | BadNets | IA | Clean Label | WaNet | Benign |
|--------|---------|------|-------------|-------|--------|
| AE | 3.25 | 3.19 | 0.72 | 1.31 | n/a |
| RE | 2.61 | 2.33 | 1.82 | 2.32 | 2.35 |

**Effectiveness of backdoor attacks on DRMs:** We use attack error (AE) to evaluate the effectiveness of backdoor attacks on DRMs. Given a set of poisoned inputs, we define AE as the average regression error calculated from the output vectors and the target vector over all the poisoned inputs. AE can be regarded as the counterpart to the attack success rate for backdoor attacks on deep classification models. Besides, we define RE as the average regression error calculated from the output vectors and the correct annotations over a benign dataset. For each attack, we apply the corresponding trigger function to each image in the benign dataset $\mathcal{D}_{be}$ to generate a set of poisoned images $\mathcal{PD}_{te}$. We report AE over $\mathcal{PD}_{te}$ and RE over $\mathcal{D}_{be}$ to show the effectiveness of the above mentioned backdoor attacks on DRMs in Table 10. The experimental results show that the AE of different backdoored DRMs are pretty low, which are almost similar to the RE of these backdoored DRMs. Also, the RE of these backdoored DRMs are similar to that of the benign DRM. This proves the effectiveness of these backdoor attacks on DRMs.

### A.4 DETAILS OF BASELINE DEFENSES

We generalize two state-of-the-art defenses, i.e., Neural Cleanse (NC) (Wang et al., 2019) and FeatureRE (Wang et al., 2022b), to regression models. Since the output of DRM is defined in the continuous space, we generalize them from classification models to regression models by taking the potential target vector $y_t$ as the optimization variable. Below, we introduce the formal definitions of the optimization problems for these two defenses after generalization.

**Generalization of NC:** We generalize the optimization problem defined in NC to DRMs as:

$$m^*, \Delta^*, y_t^* = \min_{m, \Delta, y_t} \frac{1}{N} \sum_{i=1}^{N} \ell(y_t, f(A(x_i, m, \Delta))) + \lambda \cdot |m|, \tag{8}$$

where $\lambda$ is the weight for the second objective; $A(\cdot)$ is a function that applies a trigger represented by $m$ and $\Delta$ to the benign image $x$; and $\ell(\cdot)$ is the loss function for training the DRMs. Specifically, $\ell(\cdot)$ is $\ell_1$ loss for gaze estimation and head pose estimation. The detailed description of $A(\cdot)$ can be found in (Wang et al., 2019).

**Generalization of FeatureRE:** We generalize the optimization problem defined in FeatureRE to DRMs as:

$$\theta^*, m^*, y_t^* = \min_{\theta, m, y_t} \ell(H((1 - m) \odot a + m \odot t), y_t) + \frac{\lambda_3}{N} \sum_{i=1}^{N} \|G_\theta(x_i) - x_i\|,$$

$$\text{where } t = mean(m \odot F(G_\theta(\{x_i\}_{i=1}^N))), a = F(\{x_i\}_{i=1}^N), \tag{9}$$

$$\text{s.t. } std(m \odot F(G_\theta(\{x_i\}_{i=1}^N))) \leq \tau_1, \|m\| \leq \tau_2,$$

where $m$ is the feature-space mask; $\lambda_3$ is the weight for the second objective; $\tau_1$ and $\tau_2$ are the thresholds for two constraints, respectively.

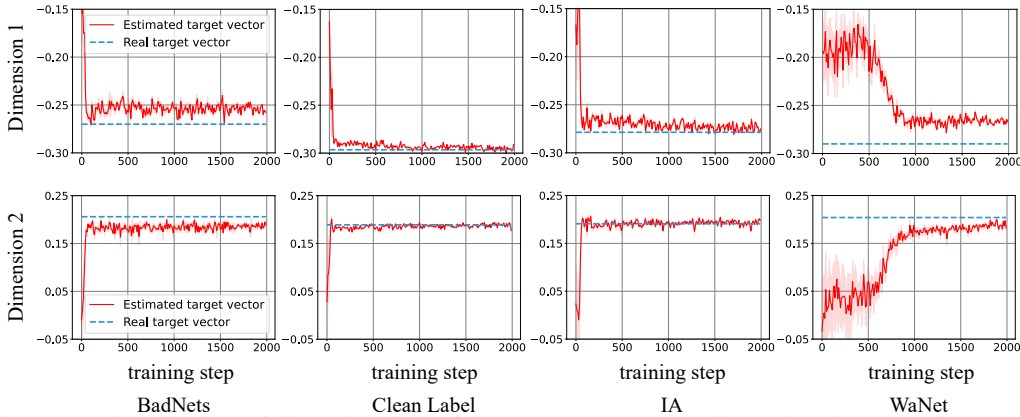

Figure 4: Visualization of the estimation of the target vector (two-dimensional vector) for DRMs backdoored by different attacks on MPIIFaceGaze dataset during the reverse engineering process. The two rows correspond to the first and second dimensions of the output of the DRMs. The red curves denote the estimation of the corresponding dimension of the target vector, while the blue curves denote that of the real target vector. The red curves can converge to the neighbor of the blue curves, which means that DRMGuard can estimate the target vector.

## A.5 ESTIMATION OF THE TARGET VECTOR

DRMGuard is able to estimate the target vector. Given a set of reversed poisoned images, we use the mean vector of the corresponding output vectors as the estimation of the target vector. We visualize the estimation of the target vector during the reverse engineering process for different backdoor attacks, i.e., BadNets, Clean Label, IA, and WaNet, on MPIIFaceGaze dataset in Figure 4. Note that, since it is infeasible to train a backdoored DRM $f$ that precisely outputs the attacker-chosen target vector for poisoned images, we use the vector, $\mathbb{E}_{x \in \mathcal{D}_{be}} f(\mathcal{A}(x))$, as the *real target vector*, where $\mathbb{E}(\cdot)$ is a function to obtain the mean vector for a given set of vectors. Figure 4 shows that the output vectors of the reversed poisoned images can converge to the neighbor of the real target vector.

## A.6 VISUALIZATION OF REVERSED POISONED IMAGES

To investigate if DRMGuard can reverse engineer the poisoned images, we show the benign images, the original poisoned images generated by different backdoor attacks, and the reversed poisoned images in Figure 5. Specifically, we randomly sample six benign images from the benign dataset $\mathcal{D}_{be}$ and show them in Figure 5(a). We then show the original poisoned images generated by BadNets, Clean Label, IA, and WaNet, on MPIIFaceGaze dataset and the corresponding reversed poisoned images in Figure 5(b), Figure 5(c), Figure 5(d), and Figure 5(e), respectively. DRMGuard is able to reverse engineer the poisoned images that are close to the original poisoned images.

## A.7 MULTI-TARGET ATTACK

We consider the scenario that the DRM is backdoored by multiple trigger functions with different target vectors. We generalize WaNet such that it can insert two warping-based triggers into a DRM. Specifically, we generate two different warping functions, and each of them corresponds to a unique target vector. During the training process, we randomly select 15% images in the mini-

Table 11: Backdoor identification performance of DRMGuard for multi-target WaNet.

| TP | FP | FN | TN | Acc |
|----|----|----|----|-----|
| 10 | 1  | 0  | 9  | 95% |

batch to apply the first warping function and change their annotations to the corresponding target vector. We then select another 15% images in the minibatch to apply the second warping function and change their annotations to the corresponding target vector. We do not modify other procedures of WaNet. We call the generalized WaNet *multi-target WaNet*. We train ten backdoored DRMs by using multi-target WaNet and ten benign DRMs on MPIIFaceGaze dataset. We report the performance of backdoor identification of DRMGuard in this scenario in Table 11, which shows that our method is effective on identifying DRMs with multiple backdoors.

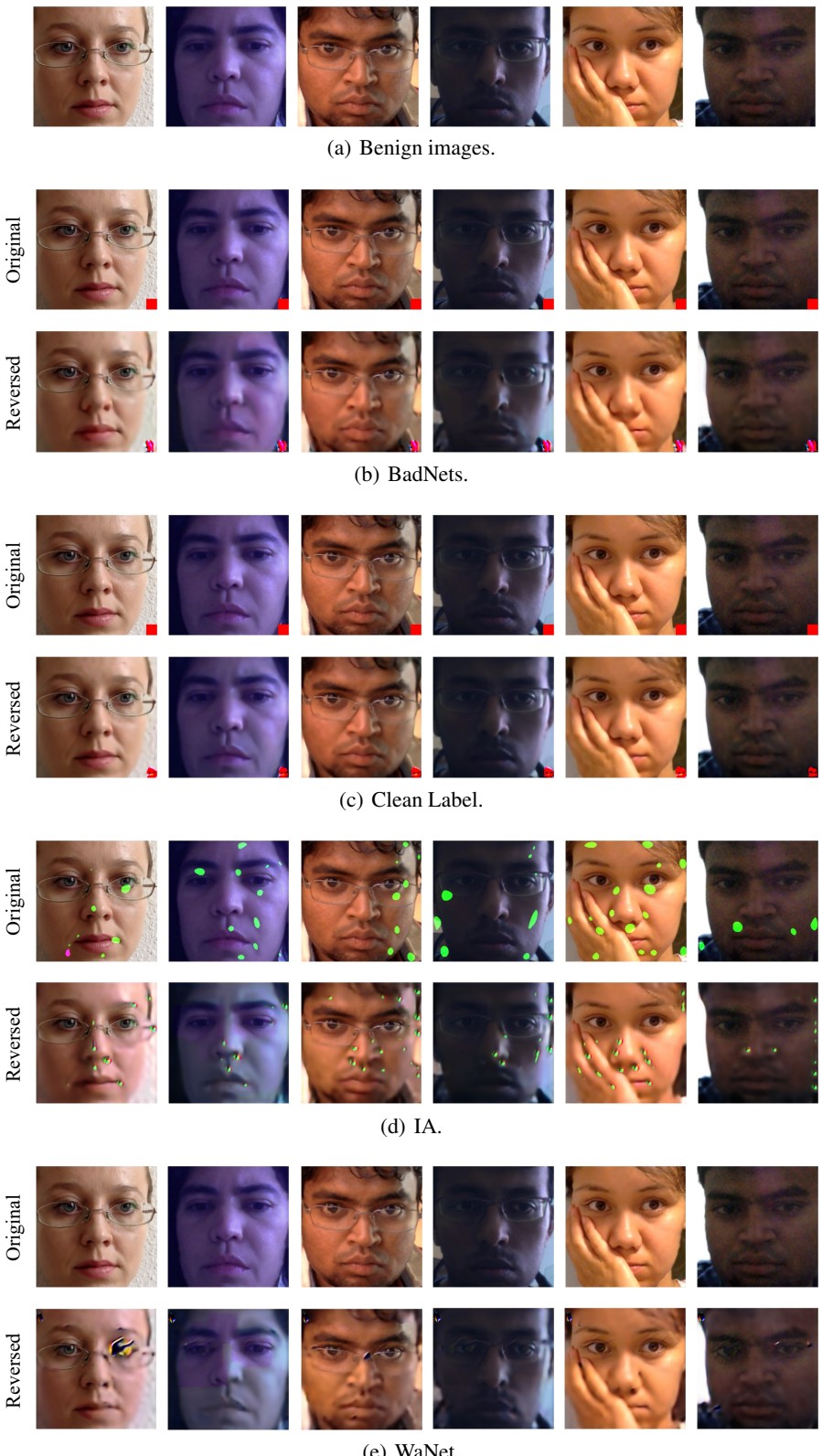

Figure 5: Comparison between (a) the benign images, and the original poisoned images and the corresponding reversed poisoned images for (b) BadNets, (c) Clean Label, (d) IA, and (e) WaNet. The reversed poisoned images are close to the original poisoned images.

Table 12: Backdoor identification performance of DRMGuard for different architectures.

| Architecture | TP | FP | FN | TN | Acc |
|---|---|---|---|---|---|
| ResNet18 | 10 | 1 | 0 | 9 | 95% |
| ResNet34 | 10 | 0 | 0 | 10 | 100% |
| MobileNetV1 | 7 | 0 | 3 | 10 | 85% |

Table 13: Backdoor mitigation performance of DRMGuard for different architectures.

| Architecture | Undefended | | DRMGuard | |
|---|---|---|---|---|
| | AE | DAE | AE | DAE |
| ResNet18 | 1.31 | 15.90 | 15.36 | 3.29 |
| ResNet34 | 2.82 | 15.97 | 15.50 | 2.40 |
| MobileNetV1 | 0.42 | 16.04 | 15.88 | 3.70 |

Table 14: Backdoor identification performance of DRMGuard on more attacks.

| Attack | TP | FP | FN | TN | Acc |
|---|---|---|---|---|---|
| Blend | 10 | 1 | 0 | 9 | 95% |
| SIG | 10 | 1 | 0 | 9 | 95% |

Table 15: Backdoor mitigation performance of DRMGuard on more attacks.

| Attacks | Undefended | | DRMGuard | |
|---|---|---|---|---|
| | AE | DAE | AE | DAE |
| Blend | 2.94 | 14.51 | 12.82 | 4.56 |
| SIG | 1.83 | 15.96 | 16.11 | 2.93 |

## A.8 More Experiments

**Evaluation on more architectures:** We study the generalization ability of DRMGuard on different architectures of DRMs. Specifically, beyond ResNet18, we consider two different architectures, i.e., ResNet34 (He et al., 2016) and MobileNetV1 (Howard et al., 2017). For each architecture, we train ten backdoored DRMs by WaNet and ten benign DRMs on MPIIFaceGaze dataset. We report backdoor identification performance and backdoor mitigation performance for different architectures in Table 12 and Table 13, respectively. The results show that DRMGuard can be generalized to different architectures of DRMs.

**Evaluation on more attacks:** We further evaluate the performance of DRMGuard on defending Blend attack (Barni et al., 2019) and SIG (Liu et al., 2020). For each attack, we train ten backdoored DRMs by WaNet and ten benign DRMs on MPIIFaceGaze dataset. We report backdoor identification performance in Table 14. We also report backdoor mitigation performance in Table 15. The experimental results show that DRMGuard can consistently defend against various attacks.

**Evaluation on a larger set of DRMs:** We evaluate the backdoor identification performance of DRMGuard on a larger set of DRMs. Specifically, we train 30 backdoored DRMs by WaNet and 30 benign DRMs on MPIIFaceGaze dataset. We report backdoor identification performance in Table 16. The results show DRMGuard can reach the identification accuracy at 90% on a larger set of DRMs.

Table 16: Backdoor identification performance of DRMGuard on a larger set of DRMs.

| TP | FP | FN | TN | Acc |
|---|---|---|---|---|
| 29 | 5 | 1 | 25 | 90% |

## A.9 Generalization to Classification Domain

We show that the idea of *minimizing the variance in the output space* can be extended to DCMs. By doing so, we can identify backdoored DCMs without enumerating all the labels.

We use $\{S(C(A(x_i)))\}_{i=1}^N$ to denote the probability vectors obtained by a backdoored DCM $C(\cdot)$ from a set of poisoned images $\{\mathcal{A}(x_i)\}_{i=1}^N$, where $S(\cdot)$ is the softmax function. Intuitively, the poisoned images will lead to similar probability vectors for a backdoored DCM. Therefore, $\frac{1}{d} \sum_{j=1}^d \sigma^2(\{S_j(C(A(x_i)))\}_{i=1}^N)$ will be a small positive value, where $S_j(C(A(x_i)))$ is the $j$th component of $S(C(A(x_i)))$. Based on this intuition, we propose the following optimization objective to generalize DRMGuard from the regression domain to the classification domain.

$$\theta^* = \min_\theta \frac{\lambda_1}{d} \sum_{j=1}^d \sigma^2\left(\{S_j(f(G_\theta(x_i)))\}_{i=1}^N\right) + \frac{1}{N} \sum_{i=1}^N \|G_\theta(x_i) - x_i\|_1 \qquad (10)$$

The first objective in the optimization problem aims to reverse engineer the poisoned images $\{G_\theta(x_i)\}_{i=1}^N$ that lead to the similar probability vectors, regardless of their actual contents. The second optimization term ensures the transformed images $G_\theta(x_i)$ is similar to original image $x_i$. We remove the feature-

Table 17: Backdoor identification performance for DCMs.

| TP | FP | FN | TN | Acc |
|---|---|---|---|---|
| 7 | 0 | 3 | 10 | 85% |

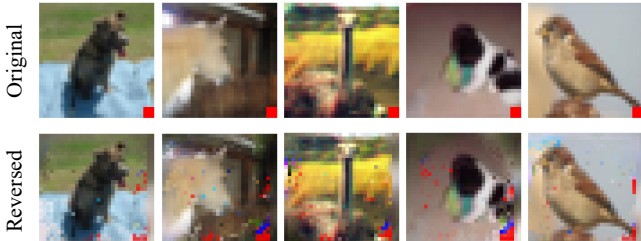

Figure 6: Illustration of original poisoned images (the first row) and reversed poisoned image (the second row) for backdoored DCMs,

space regularization term designed for DRMs, since backdoored DRMs and backdoored DCMs have different feature-space characteristics.

We train ten backdoored DCMs by BadNets and ten benign DCMs on Cifar10. We report backdoor identification results in the Table 17. We visualize the reversed poisoned image in Figure 6. The results show that our idea of minimizing the variance in the output space can be extended to the classification domain for backdoor identification without enumerating all the labels.

### A.10 TECHNICAL DETAILS OF DRMGUARD

We use a simple generative model to implement $G_\theta$, which is similar to the generative model used in (Nguyen & Tran, 2020). Before performing the reverse engineering, we pre-train $G_\theta$ on the benign dataset for 5,000 training steps. During the reverse engineering, the batch size for gaze estimation datasets is 50, while that for head pose estimation datasets is 100. We use Adam optimizer to train $G_\theta$ for 2,000 training steps, taking about 16 minutes. The learning rates for gaze estimation and head pose estimation are 0.0015 and 0.0001, respectively.