# OpenReview forum: "DRMGuard: Defending Deep Regression Models against Backdoor Attacks"
_ICLR.cc/2024/Conference — Submitted to ICLR 2024_

### Official Review · Reviewer_tXTC · 2023-10-30

**Soundness:** 3 good
**Presentation:** 3 good
**Contribution:** 1 poor
**Rating:** 3
**Confidence:** 4

**Summary:**

This paper proposes a model-based backdoor defense method against the deep regression model. This paper proposes to leverage trigger reverse to detect and remove backdoors from the deep regression models. The experiments demonstrates its effectiveness.

**Strengths:**

1.This paper investigates backdoor defense in deep regression model, which has not been explored before.

**Weaknesses:**

1. The novelty is limited. The trigger reverse method is similar to Neural Cleanse.
2. This paper should emphasize the importance of the backdoor in the regression models. Why is it important in the regression area?
3. The number of categories of compared methods and backdoor attacks is clearly below ICLR acceptance threshold.

**Questions:**

Why is backdoor defense important in the regression area?

---

> ### Author Response · Authors · 2023-11-13
> **Reply by authors**
>
> We appreciate your time and comments on our submission. Please kindly find our response below.
>
> > Weakness 1: The novelty is limited. The trigger reverse method is similar to Neural Cleanse.
>
> **Response:** Thanks for your comments. **Our method is significantly different from Neural Cleanse (NC)**, as we have discussed in the third paragraph of the introduction and the third paragraph of the related work. Below, we briefly discuss the differences again.
>
> NC is designed for backdoored deep classification models (DCMs). First, different from DCMs for which the output space is discretized into a few class labels, the output space of DRMs is continuous. Thus, **it is infeasible to enumerate all the potential target vectors by NC to determine the infected target vector.**
>
> Second, different from DCMs that adopt $\arg\max$ to obtain the final output, DRMs do not need $\arg\max$. This makes the backdoor behavior of the backdoored DRMs distinct from that of the backdoored DCMs. Specifically, for a backdoored DCM, the backdoor behavior is often triggered by the activation values of several neurons in the feature space, whereas for a backdoored DRM, it is triggered by the activation values of all the neurons. **Our method introduces feature-space regularization term based on the unique feature-space characteristics of backdoored DRMs, while NC does not utilize the feature-space characteristics of backdoored DCM for reverse engineering.**
>
> Our work is the first to detect backdoored DRMs in the image domain. We apply our method to a DRM to reverse engineer a potential trigger function, based on which we decide whether the model has been injected with a backdoor or not. A major challenge to reverse engineering the potential trigger function in the regression domain is that the output is defined in the continuous space. To address this challenge, we formulate reverse engineering as an optimization problem, which is based on both unique output-space and feature-space characteristics of the backdoored DRMs that are observed in this paper.
>
> These differences make our work distinct from NC as well as all existing defenses that are designed for backdoored DCMs.
>
> > Weakness 2 and questions: This paper should emphasize the importance of the backdoor in the regression models. Why is it important in the regression area?
>
> **Response:** Thanks for your comments. We have discussed the motivation and importance of defending deep regression models against backdoor attacks in the first paragraph of the introduction section and the first paragraph of the evaluation section. Below, we briefly summarize the importance again.
>
> **Deep regression models are used in many safety-critical applications**, such as driver attention monitoring, head pose estimation, and facial landmark detection, among many others. In our evaluation, we consider gaze estimation and head pose estimation as two representative applications. Specifically, gaze estimation tracks where the subject is looking, and plays a key role in a series of safety-critical applications, such as user authentication, driver distraction detection, and user attention monitoring in AR/VR. Similarly, head pose estimation has also been used in many daily applications, such as the driver assistance system, pedestrian attention monitoring, and aliveness detection in facial recognition.
>
> Unfortunately, DRMs are also vulnerable to backdoor attacks, and existing defenses designed for classification models are ineffective in detecting backdoored regression models. **This research problem has been largely overlooked by the community and we are the first to detect backdoored DRMs in the image domain.**
>
> > Weakness 3: The number of categories of compared methods and backdoor attacks is clearly below ICLR acceptance threshold.
>
> **Response:** Thanks for your comments. Since **there is no existing defense designed for DRMs in the image domain**, we selected the most relevant and representative defenses originally designed for DCMs and adapted them to the regression problem for comparative analysis. Specifically, we compare DRMGuard with four compared methods, i.e., Neural Cleanse, FeatureRE [1], Fine-pruning [2], and ANP [3], and show that DRMGuard can defend against six backdoor attacks.
>
> [1] Zhenting Wang, et al. "Rethinking the Reverse-engineering of Trojan Triggers." NeurIPS 2022.
>
> [2] Kang Liu, et al. "Fine-pruning: Defending against backdooring attacks on deep neural networks." International symposium on research in attacks, intrusions, and defenses. Cham: Springer International Publishing, 2018.
>
> [3] Dongxian Wu, and Yisen Wang. "Adversarial neuron pruning purifies backdoored deep models." NeurIPS 2021.

---

> > ### Author Response · Authors · 2023-11-14
> > **Invitation for Further Feedback and Guidance on Our Response to Your Comments**
> >
> > We value the comments and feedback you provided. Please kindly take a look at our response and give us further instructions on how we can address your comments and improve our work.

---

> ### Author Response · Authors · 2023-11-20
> **Invitation for More In-Depth Feedback**
>
> Dear Reviewer tXTC,
>
> Thank you for your valuable questions and comments. We're eager to continue this dialogue and are prepared to respond to any additional queries you may have. In light of our forthcoming responses, we kindly ask if you would be willing to reevaluate your score, should our answers help to alleviate your concerns. Your feedback is greatly appreciated.
>
> Sincerely,
>
> The authors

---

### Official Review · Reviewer_FMPV · 2023-10-31

**Soundness:** 1 poor
**Presentation:** 2 fair
**Contribution:** 2 fair
**Rating:** 3
**Confidence:** 5

**Summary:**

This paper proposed a model-level backdoor defense which tries to reverse-engine the trigger signal from backdoored model. Specifically, the authors considered a new scenario, i.e., deep regression task, where the output of the model is a vector instead of discrete output in the deep classification task. For instance, in a deep regression task, called gaze estimation, the deep regression model will output a vector to represent the direction of one person view. In this new scenario, the existing related work will not work since some of them [(Wang et al., 2019] recovered a suspect trigger signal for each class, and then determine the true trigger from all suspects. However, the existing works cannot be directly used in deep regression tasks since the defender cannot the infinite outputs. The main contribution of this work solves this problem.

**Strengths:**

The authors proposed new reverse engineering method to recover the backdoor attack in the deep regression task.

**Weaknesses:**

The main idea to reconstruct the trigger is based on the generative model, which has been exploited following references:
[1] Zhu, Liuwan, et al. "Gangsweep: Sweep out neural backdoors by gan." Proceedings of the 28th ACM International Conference on Multimedia. 2020.
[2] Chen, Huili, et al. "DeepInspect: A Black-box Trojan Detection and Mitigation Framework for Deep Neural Networks." IJCAI. Vol. 2. No. 5. 2019

Secondly, the new proposed method proposed a new regularization, which is $r_{f}$ shown in Equation 5. This regulation is only designed based on the empirical results, i.e., the angle of poisoned inputs is more concentrated than the benign data. There is not any theoretical proof to support this.

**Questions:**

Is it possible to compare the ABS method, which is only mentioned in the introduction but isn’t as a state-of-the-art to compare. The reason why I raise this question is that the ABS directly analyze the middle layer feature to reconstruct the trigger. It is will not affected by the infinite of output influence made by deep regression task.

---

> ### Author Response · Authors · 2023-11-13
> **Reply by authors**
>
> We appreciate your time and comments on our submission. Please kindly find our response below.
>
> > Weakness 1: The main idea to reconstruct the trigger is based on the generative model, which has been exploited following references: [1] Zhu, Liuwan, et al. "Gangsweep: Sweep out neural backdoors by gan." Proceedings of the 28th ACM International Conference on Multimedia. 2020. [2] Chen, Huili, et al. "DeepInspect: A Black-box Trojan Detection and Mitigation Framework for Deep Neural Networks." IJCAI. Vol. 2. No. 5. 2019
>
> **Response:** Thanks for your comments. The papers mentioned by the reviewer use generative models to perform trigger reverse engineering for classification models and are required to enumerate all possible labels. By contrast, our work is distinct from these works in the following ways.
>
> First, as the output vector of deep regression models (DRMs) is defined in the continuous space, it contains countless potential target vectors. **We minimize the variance of the output vectors to perform reverse engineering for backdoored DRMs without enumerating each potential target vector.**
>
> Second, we analyze the difference in feature space between backdoored DRMs and backdoored deep classification models (DCMs), and identify the unique feature-space characteristics of backdoored DRMs by performing experiments and theoretical analysis. **Thus, the second major difference in our work is the observation of the unique feature-space characteristics of backdoored DRMs and the feature-space regularization term for reverse engineering based on this observation.**
>
> > Weakness 2: Secondly, the newly proposed method proposed a new regularization, which is $r_f$ shown in Equation 5. This regulation is only designed based on the empirical results, i.e., the angle of poisoned inputs is more concentrated than the benign data. There is not any theoretical proof to support this.
>
> **Response:** Thanks for your comments. In addition to the empirical results, **in Section 3.3 of the original submission, we have provided a comprehensive theoretical analysis to support the observation of the feature-space characteristics**. Please see the *Theoretical analysis and metrics* and *the last paragraph* of Section 3.3.
>
> > Questions: Is it possible to compare the ABS method, which is only mentioned in the introduction but isn’t as state-of-the-art to compare? The reason why I raise this question is that the ABS directly analyze the middle layer feature to reconstruct the trigger. It is will not affected by the infinite of output influence made by deep regression task.
>
> **Response:** Thanks for your comments. Although ABS is not affected by the infinite output of DRMs, it is built upon the observation that successful backdoor attacks will entail one or a set of inner neurons, which will fall within a certain range to predict the target labels (see Section 3.2 of the ABS paper). This observation is similar to that in both Fine-pruning [1] and ANP [2]. As we have discussed in Section 3.3 (see *the difference between backdoored DCM and DRM* in Section 3.3), this observation does not hold for backdoored DRM. **We also conduct experiments in Section 4.3, which confirms our analysis that existing defenses built upon the feature-space characteristics of backdoored DCM cannot be applied to backdoored DRMs.**
>
> [1] Kang Liu, et al. "Fine-pruning: Defending against backdooring attacks on deep neural networks." International symposium on research in attacks, intrusions, and defenses. Cham: Springer International Publishing, 2018.
>
> [2] Dongxian Wu, and Yisen Wang. "Adversarial neuron pruning purifies backdoored deep models." NeurIPS 2021.

---

> > ### Author Response · Authors · 2023-11-14
> > **Invitation for Further Feedback and Guidance on Our Response to Your Comments**
> >
> > Thank you again for your comments. We kindly invite you to review our response and provide further guidance on how we can address your points and enhance our work. Your time and effort will be deeply appreciated.

---

> > > ### Author Response · Authors · 2023-11-20
> > > **Invitation for More In-Depth Feedback**
> > >
> > > Dear Reviewer FMPV,
> > >
> > > Thank you for your valuable questions and comments. We're eager to continue this dialogue and are prepared to respond to any additional queries you may have. In light of our forthcoming responses, we kindly ask if you would be willing to reevaluate your score, should our answers help to alleviate your concerns. Your feedback is greatly appreciated.
> > >
> > > Sincerely,
> > >
> > > The authors

---

### Official Review · Reviewer_WvrB · 2023-11-03

**Soundness:** 3 good
**Presentation:** 3 good
**Contribution:** 3 good
**Rating:** 8
**Confidence:** 5

**Summary:**

The authors propose a backdoor defense method named DRMGuard to tackle backdoor attacks of deep regression models. The core technique is to optimize the variance of feature representations when the cadidate outputs are uncountable. The authors conduct extensive experiments to demonstrate the effectiveness of DRMGuard.

**Strengths:**

1.	The research problem is of great significance
2.	The paper is well-structured and easy to follow.
3.	The idea is novel and inspiring.

**Weaknesses:**

1.	Time complexity. I notice that the authors did not report the running time of DRMGuard. I am not sure whether the variance calculation is time-consuming and could be scaled to higher dimensional conditions.
2.	Extension to classification models (not very important). In my opinion, the key of this work is that we could use the variance of deep representations for trigger inverse optimization when the candidate labels are uncountable. I think this method could be extended to deep classification models. I suggest the authors conducting this method in classification experiments.
3.	Typos. In Table 8, MRT -> MTR.

Overall, I like this paper. I think the authors study an important problem. Besides, the idea of optimizating the variance of deep representations will inspire more future works.

**Questions:**

Please refer to Weaknesses.

---

> ### Author Response · Authors · 2023-11-13
> **Reply by authors**
>
> We really appreciate your constructive comments and insightful suggestions. Please kindly find our response below.
>
> > Time complexity. I notice that the authors did not report the running time of DRMGuard. I am not sure whether the variance calculation is time-consuming and could be scaled to higher dimensional conditions.
>
> **Response:** Thanks for your constructive comment. To investigate the processing time of our method, we perform experiments on a server installed with an NVIDIA A10 GPU. The input size for the generative model $G_{\theta}$ and the deep regression model (DRM) is $224\times 224 \times3$. The generative model consists of 14 convolutional layers, and the DRM is implemented by ResNet18. The batch size is 50. **Given a DRM, DRMGuard takes about 16 minutes to train $G_{\theta}$ for 2000 training steps.** We also measure the processing time in calculating the variance when the output is a 1000-dimensional vector. We repeat this process 1000 times, and **the average time used to calculate the variance over 50 1000-dimensional vectors is $5.91\times 10^{-2}$ ms**. We think the results show that DRMGuard can be scaled to higher dimensional conditions.
>
> > Extension to classification models (not very important). In my opinion, the key of this work is that we could use the variance of deep representations for trigger inverse optimization when the candidate labels are uncountable. I think this method could be extended to deep classification models. I suggest the authors conducting this method in classification experiments.
>
> **Response:** This is a very insightful suggestion, and we really appreciate it. We analyze the behaviors of backdoored deep classification models (DCMs) and design a prototype to show that the idea can also be extended to DCMs. By doing so, we can perform trigger inverse optimization for backdoored DCMs without enumerating all the labels.
>
> Specifically, we train a backdoored DCM on Cifar10 by BadNet. We use  $\lbrace x_i \rbrace _{i=1}^N$ and $\lbrace\mathcal{A}(x_i)\rbrace _{i=1}^N$ to denote a set of benign images and poisoned images, respectively, where $\mathcal{A}(\cdot)$ is the function to apply the trigger to the image. We feed $\lbrace x_i \rbrace _{i=1}^N$ and $\lbrace\mathcal{A}(x_i)\rbrace _{i=1}^N$ to the backdoored DCM $C(\cdot)$, followed by a softmax function $S(\cdot)$ to obtain the probability vectors $\lbrace S(C(x_i))\rbrace _{i=1}^N$ and $\lbrace S(C(A(x_i)))\rbrace _{i=1}^N$. We observe that poisoned images lead to similar probability vectors for the backdoored DCM. Formally, we have the following observation: $\frac{1}{d}\sum _{j=1}^d \sigma^2(\lbrace S_j(C(A(x_i))) \rbrace _{i=1}^N) << \frac{1}{d}\sum _{j=1}^d \sigma^2(\lbrace S_j(C(x_i)) \rbrace _{i=1}^N)$, where $S_j(C(A(x_i)))$ and $S_j(C(x_i))$ is the $j$th component of $S(C(A(x_i)))$ and $S(C(x_i))$ respectively. Specifically, we observe that $\frac{1}{d}\sum _{j=1}^d \sigma^2(\lbrace S_j(C(A(x_i))) \rbrace _{i=1}^N) \approx 9\times 10^{-12}$  and $\frac{1}{d}\sum _{j=1}^d \sigma^2(\lbrace S_j(C(x_i)) \rbrace _{i=1}^N) \approx7\times 10^{-2}$.
>
> Based on this observation, we propose the following optimization objective to generalize DRMGuard from the regression domain to the classification domain:
> $$
> \theta^{*}=\min_{\theta} \frac{\lambda_1}{d}\sum _{j=1}^d {\sigma^2(\lbrace S_j(C(G _{\theta}(x_i))) \rbrace _{i=1}^N)} + \frac{1}{N} \sum _{i=1}^N \| G _{\theta}(x_i)-x_i \|_1
> $$
>
> The first objective in the optimization problem aims to reverse engineer the poisoned images $\lbrace G _{\theta}(x_i)\rbrace _{i=1}^N$ that lead to similar probability vectors, regardless of their actual contents. The second optimization term ensures the transformed image  $G _{\theta}(x_i)$ is similar to the original image $x_i$. We remove the feature-space regularization term designed for DRMs, since backdoored DRMs and backdoored DCMs have different feature-space characteristics, as we discussed in our paper.
>
> By solving the above optimization problem, $G_{\theta}$ can reverse engineer the poisoned images without enumerating each label. Also, the probability vectors can converge to the vector that leads to the target label. We train ten backdoored DCMs by BadNets and ten benign DCMs on Cifar10. We report backdoor identification results in the following table.
> | TP | FP | FN | TN | Acc |
> |:--:|:--:|:--:|:--:|:---:|
> |  7 |  0 |  3 | 10 | 85% |
>
> **We also show reversed poisoned images in Appendix A.9 of the updated supplementary.** We think the results show that our idea of minimizing the variance can be extended to the classification domain for reverse engineering.
>
> >Typos. In Table 8, MRT -MTR.
>
> **Response:** Thanks for your detailed comments. We have fixed the typos in the updated version.

---

> > ### Comment · Reviewer_WvrB · 2023-11-13
> >
> > Thanks for your detailed response, which has addressed all my concerns. I appreciate this paper and have updated my confidence score. Good luck !

---

> > > ### Author Response · Authors · 2023-11-13
> > > **Reply by authors**
> > >
> > > We greatly appreciate your time and efforts in reviewing our submission. Thank you very much for your insightful comments and support.

---

### Meta-Review · Area_Chair_kUAy · 2023-12-09

**Metareview:**

This work studies the defense of backdoor attack against deep regression models.
It investigated the difference between regression and classification models, and found some special characteristics in backdoored regression models, from the perspective of norm variance and angle variance in the feature space. Based on the finding, a reverse engineering based defense was proposed.

Most reviewers recognized the value of exploring the backdoor of a rarely studied model, and the finding of variance is recognized by some reviewers (including me). However, several important concerns were also proposed, mainly including:
1. Justification of the importance of the new task. The authors provided some explanations in the rebuttal about the importance of defending deep regression models. But it is a bit insufficient. It is expected to show stronger justification when starting a new task.
2. Technical novelties: in terms of the novelty of the proposed defense method, although the authors claimed the special design for regression models, it is obvious that the method, even the finding of variance, has been inspired by existing works. I think that is ok, but the authors should clearly demonstrate the inspiration and difference.
3. Insufficient comparisons. The authors claimed that there are no existing works specially designed regression models, thus they just compared a few. It is not very convincing to me. It is trivial to extend most backdoor attack/defense methods from classification to regression models, such as most pruning or fine-tuning based methods, as well as the backdoor model detection methods utilizing the features or weights. If existing methods work well for the new task, the value of new specially designed method is doubtable. The authors should demonstrate the drawback/limitation of existing works on new tasks, then started to develop new methods.

In summary, although the studied task is valuable, and there are some interesting findings, there are still large space to make it more solid. Hope all reviews are helpful to further enhance this work.

**Justification For Why Not Higher Score:**

new task, a few new finding, but lack of solid justification and sufficient comparisons.

**Justification For Why Not Lower Score:**

n/a

---

### Decision · Program_Chairs · 2024-01-16

Reject